# Optimal Surface Pre-Reacted Glass Filler Ratio in a Dental Varnish Effective for Inhibition of Biofilm-Induced Root Dentin Demineralization

**DOI:** 10.3390/polym14225015

**Published:** 2022-11-18

**Authors:** Syed Ali Murtuza, Khairul Matin, Noriko Hiraishi, Yasushi Shimada

**Affiliations:** 1Department of Cariology and Operative Dentistry, Division of Oral Health Sciences, Graduate School of Medical and Dental Sciences, Tokyo Medical and Dental University (TMDU), Tokyo 113-8549, Japan; 2Department of Oral Medicine and Stomatology, School of Dental Medicine, Tsurumi University, Tsurumi-ku, Yokohama City 230-8501, Japan; 3Medoc International Co., Ltd., Tokyo 102-0094, Japan

**Keywords:** S-PRG filler, dental varnish, cariogenic biofilm, oral biofilm reactor, confocal laser scanning microscope, swept-source optical coherence tomography, SEM-EDS

## Abstract

A unique type of dental varnish (DV) containing surface pre-reacted glass (S-PRG) fillers of different concentrations was evaluated to determine the unpresented optimal ratio for inhibiting root dentin bio-demineralization. S-PRG DVs (10% to 40%)—*10%-S*, *20%-S*, *30%-S*, and *40%-S*—were applied to bovine root dentin blocks and compared with controls—*0%-f* (no S-PRG) and *5%-NaF* (5%-NaF). The *Streptococcus mutans* biofilm challenge was executed inside and outside an oral biofilm reactor for 7 days. The specimens were examined using a confocal laser scanning microscope and swept-source optical coherence tomography. Furthermore, they were observed using a scanning electron microscope and analyzed using energy-dispersive X-ray spectroscopy. The roughness (SzJIS) due to leaching of DV materials and demineralization depth were significantly less in the S-PRG groups than the control groups (*p* < 0.05). Complete or partially plugged dentinal tubules (DTs) were observed in *20%-S*, *30%-S*, and
*40%-S*, while wide-open DTs were observed more in controls. Cylindrical tags were present in groups containing more than 20% S-PRG. F, Na, Al, and Sr were detected in a higher percentile ratio in the *20%-S*, *30%-S*, and *40%-S* groups compared to *0%-f* and *10%-S*. Nonetheless, it is suggested that incorporating 20% to 30% S-PRG fillers in DVs would be effective enough as an anti-demineralization coating, together with supplementing minerals; further evaluation is required to validate these findings.

## 1. Introduction

Varnishes, in general, are defined by an adherent coating material that forms a thin layer on surfaces after application, creating a solid film after physical or chemical changes, working as a functional coating [1]. Although varnishes are more commonly recommended for children, they are also considered for any individual with a high risk of caries [2]. The properties of the varnishes are related to the prevention of demineralization [3,4,5] and the treatment of hypersensitivity [6]. According to the U.S. Patent Application claims priority to U.S. Provisional Application No. 61/765,062, filed on 15 February 2013, a DV is generally comprised of natural gum rosin, resin, NaF, and various solvents, flavor additives, sweetener, and pigments with fluoride-releasing properties. Newer varnishes have a much more complex composition, with the addition of filler content. The addition of filler plays a role in providing a better structural shape and consistency with the distribution of filler particles [7].

The effects of fluoride-releasing varnishes on enamel, dentin, and root caries have been reported in papers and discussed in a review paper involving several articles [8,9]. Fluoride uptake, reaction, and release are strongly dependent on the duration of contact [10,11]. In vitro and in vivo studies have shown that varnishes supply fluoride more efficiently than other topical agents, with reductions in caries ranging from 50 to 70% [5,12,13]. However, studies on the general health and dental effects, including fluorosis and the safety of fluoride [14], described that the fluoride-releasing ability alone is not enough for inhibiting bacterial invasion, which usually leads to secondary caries [15]. 

Recently, a new series of dental varnishes (DVs; Shofu Inc., Kyoto, Japan) was developed with different percentages of S-PRG filler particles, which has a three-layer bioactive filler material produced by pre-reactant glass (PRG) technology, where a glass ionomer reaction (acid–base reaction) between surface-treated multi-functional glass (fluoroboroaluminosilicate glass) filler and the polyacrylic acid aqueous solution is employed to preliminary form a stable glass ionomer phase on the surface of the glass, which has been applied to various dental materials. The concept of incorporating S-PRG fillers with multi-ion-releasing properties in dental materials has been a focal area for the last few years. The pre-reacted glass ionomer phase on the surface of the glass core of the S-PRG filler can release multiple ions, such as Na^+^ (Na), Sr ^2+^ (Sr), Al ^3+^ (Al), F^-^ (F), BO_3_^3−^ (B), and SiO_3_^2−^ (Si), which impart various properties to the varnishes and have the ability to release multiple ions. They should have more potential for long-term durability and self-repair abilities against demineralization at the dentin–enamel interface [16,17,18,19].

*Streptococcus mutans (S. mutans)* has been implicated as a primary causative agent of dental caries in humans, and one of its important virulence properties is an ability to form a biofilm known as dental plaque on the tooth surface [20]. Material properties may impact the growth of microorganisms that metabolize sugar and create an acidic environment where aciduric bacteria (including *mutans streptococci*) become dominant, which leads to the demineralization of enamel and dentin [21]. In reality, most microorganisms exist in biofilms, a community of many of them from the same species or other genera, and are attached to surfaces [21]. The prevention of the adherence of cariogenic bacteria to tooth surfaces is considered to be an important strategy for controlling dental caries [22]. Furthermore, a custom-built in vitro system was mentioned in previous studies for the formation of cariogenic biofilm through the use of a computer-controlled oral biofilm reactor (OBR) for the study of its material properties and its impact on prevention of caries [23].

To date, no approach for determining the optimal filler ratio of S-PRG in DVs methodically applied to root dentin has been reported. Hence, evaluating the new DVs introduced with S-PRG fillers to determine the optimal filler ratio effective for protecting root dentin from cariogenic biofilm was considered in this in vitro study. Therefore, it aimed to investigate the efficacy of the experimental DVs on inhibition of bovine root dentin demineralization, to determine the optimal filler ratio adequate for protecting root dentin from cariogenic biofilm primarily forming in the OBR, and to assess the plugging conditions of DTs, speculating clinical hypersensitivity. 

We hypothesized that: (1) the new S-PRG filler containing DV concentrations will dependently show a significantly higher anti-demineralization effect on root dentin in comparison to DVs without S-PRG fillers; and (2) the higher the concentration of S-PRG fillers in the DVs, the better the plugging of the DTs will be. 

## 2. Materials and Methods

### 2.1. Specimen Preparation

A total of 40 intact extracted bovine incisor roots were preserved at 4 °C in the refrigerator before use. Roots were separated from the crown using a diamond disc. The specimens (4 × 4 × 2 mm³) were prepared from the cervical portion of the root (Figure 1). 

The exposed root surfaces were ground flat by polishing successively with #1000-, #1200-, #1500-, and #2000-grit silicon carbide paper (SiC; DCCS, Sankyo Fuji Star, Saitama, Japan); then, the samples were divided into two sets. One set of samples was ultra-sonicated with 0.5M ethylene diamine tetra-acetic acid (EDTA) for 5 min, followed by ultrasonic wash with de-ionized water (DiW; Japan Millipore Corp., Tokyo, Japan) for 5 min. The other set was ultra-sonicated with DiW only for 5 min. The specimens were stored in a refrigerator at 4 °C inside a plastic box, maintaining moist conditions using DiW-socked wet tissue paper (renewed every day) throughout the duration of the study to prevent desiccation-induced cracks. The specimens were randomly divided into two main groups according to the materials applied and subdivided on filler contents and/or filler ratio, as shown in Table 1. 

### 2.2. Material Application

Each varnish set (base and activator) was supplied in an air-sealed transparent pack for a single application (Figure 1). In a separate box, disposable brushes were provided for mixing the base and activator before application as activated varnish.

A modified version of the basic composition mentioned by Kotoku M. et al. [24] is shown in Table 1.

Before application, the surface of the bovine tooth was cleaned with sterile DiW and dried by keeping inside a clean bench under airflow conditions for 10 min. The material seal was opened carefully. A thorough mixing was done for 2 min with the brush. The two uniform coats of material were applied using a micro-applicator brush exclusively for each material, supplied by the manufacturer, onto the prepared surface for 1 min. Soft pressure was applied using the backside of the spoon of a dental evans (a dental lab instrument with a small spoon at one end and a small knife at the other) immediately after first coating for 10~30 s in order to drive the DV materials into the open DTs. The surface was kept undisturbed for 10 min. After initial setting of material, partial scraping from one half of the surface was done with a dental knife in order to have two adjacently located parts for comparison on the same root dentin block so that the workload for specimen preparation could be reduced, inspection could be done clearly, and data could be acquired with minimum technical errors. These steps were performed inspecting under a light microscope (LM, Nikon SMZ1000, Nikon, Tokyo, Japan), particularly after the material application and after partial scraping of material from the half of the surface of the specimen to check for overhanging margins.

### 2.3. Demineralization Induced by Biofilm

A laboratory strain, *S. mutans* MT8148, was grown in brain heart infusion (BHI) broth (Becton, Dickinson and Company, MD, USA) at 37 °C under anaerobic conditions. The suspension was washed three times with phosphate-buffered solution (PBS) and stored at 4 °C with gentle stirring. For the growth of the biofilm, a solution of heart infusion broth (HI, Becton Dickinson, Sparks, MD, USA) with sucrose (1% final concentrations) was used.

The basic biofilm formation method on the specimen surfaces has been described in previously published articles [25,26]. In brief, after two-step cultures of *S. mutans* suspension, HI–sucrose medium and PBS were prepared, autoclaved, and preserved at 4 °C before being pumped into the water jacket-encircled OBR chambers and were continuously dropped for 24 h to form biofilms by maintaining the internal temperature at 37 °C. In this process, a water dome-like structure was formed by the liquids and mixed with the force of gravity exerted from falling drops before being diffusely distributed all over the specimen [27]. The pH was monitored and recorded simultaneously by the pH electrode positioned at the middle. For each experiment, 18 blocks were placed on two separate Teflon holders of OBR (nine in each of two) and repeated four times.

After 24 h incubation of the biofilm in the OBR chamber, each specimen with intact in vitro biofilm was carefully picked up from the Teflon holder and was transferred to 24-well tissue culture plates (Corning Inc. New York, NY, USA) and incubated at 37 °C in the 1.5 mL of HI medium containing 1% sucrose medium for another 6 days (Figure 1), with periodic replacement every other day. After the demineralization process, each specimen was transferred into 1 mL of 0.25-mol/L sodium hydroxide solution and shook mildly for 10 min to remove the biofilms.

### 2.4. Confocal Scanning Laser Microscope Observation

After the 7-day biofilm test, the laser penetration test for each specimen was carried out using a confocal laser scanning microscope (CLSM; VK8510, Keyence Corporation, Tokyo, Japan) at × 50 magnifications. The mean surface roughness (SzJIS; µm) of ten representative DTs for each specimen was taken and the average was plotted in a graph. The penetration depth was captured as spikes on CLSM images showing deeper valleys as the laser penetrated through the reopened DTs due to the leaching out of DVs. The images were processed by a two-line method by using an analyzer program (MultiFileAnalyzer V1.3.1.120, Keyence, Osaka, Japan). A fresh root (considered as an additional negative control) did not receive the application of any material and/or biofilm treatment.

### 2.5. Optical Coherence Tomography (SS-OCT) Observation

The focused light beam is protruded on to the selected locations and scanned across the area of interest by using an infrared scanning probe of an SS-OCT device (IVS-2000, Santec, Komaki, Japan). Tomography scanning was performed at randomly selected locations after visual inspection. The SS-OCT scanning probe was set at a fixed distance over the top surface, with the scanning beam oriented perpendicular to the lesion, and specimens were scanned in a cross-sectional direction after mildly air-drying [23]. The lesion depths of 10 positions were measured.

### 2.6. Scanning Electron Microscope (SEM) Observation

Selected specimens were sputter-coated with 30-mA gold and examined under a scanning electron microscope (JSM IT100; JEOL, Tokyo, Japan) to observe the morphological structures of the resin–tooth interface and the morphology of DVs in the DTs; they were inspected using the SEM images at 2000× magnification. For cross-sectional inspection, specimens were split at the middle with pliers after DV application specimens. One set of specimens was fixated with glutaraldehyde after biofilm formation, naturally dried for one week, and embedded in epoxy resin material (JER828, Japan Epoxy Resin Ltd., Tokyo, Japan). Epoxy resin-embedded specimens were gradually ground to expose the cross-sectional surface of DTs and polished with #2000-grit SiC. Another set of specimens was used to inspect the condition of the scraped part of the root dentin surface after 7 days of biofilm attack. All specimens were coated with a thin layer of 30 mA gold particles after drying naturally, before examining under SEM.

### 2.7. Energy Dispersive X-ray (EDS) Spectroscopy

The surfaces of the prepared specimens were examined under SEM, analyzing with an EDS of the experimental groups. The specimens which were analyzed with SEM under operating conditions of 20 kV were simulated for EDS under the conditions. Material was applied onto the carbon tape, as mentioned before, onto the stage and kept at room temperature for 7 days for moisture evaporation; these were later named material only. The surface area and point analysis were performed for the detection of boron (B), fluoride (F), sodium (Na), aluminum (Al), silica (Si), and strontium (Sr) in percentile relation to phosphorous (P) and calcium (Ca) ions.

### 2.8. Statistical Analysis

The normal distribution of data was checked. The obtained values were statistically processed using one-way ANOVA at a significance level of *p*-value < 0.5, with the Tukey HSD test for laser penetration data and demineralization evaluation for OCT study. The Shapiro–Wilk test was used to evaluate normal distribution of data. Data were analyzed by two-way ANOVA and *t*-test with Bonferroni correction at a significance level of *p*-value < 0.5 (SPSS ver. 26 for Windows, Chicago, IL, USA).

## 3. Results

Milky-white biofilms remain attached on all specimens after 24 h formation in the OBR and gradually became condensed and yellowish as the culture continued for an additional 6 days (Figure 1). The root dentin surfaces were not characteristically smooth after the removal of the biofilms when inspected under a light microscope.

### 3.1. Confocal Laser Penetration Depth Measurement

Upon the surface treatment of the specimen with 0.5 M ethylene diamine tetra-acetic acid (EDTA) before the application of the material, the images were recorded with CLSM, and the widening and clearing of the dentinal tubules were observed.

The mean depth of the CLSM laser penetration into the ten representative reopened DTs for each specimen determined by the Keyense Software MultiFileAnalyzer program, provided as surface roughness (SzJIS is the average distance between five highest peaks and five lowest valleys presented in µm) on the measurement of the two-point analysis, are shown in Figure 2 and Figure 3. The CLSM images (Figure 2) of the control group show thicker and deeper valleys (spikes) compared to the experimental groups. According to one-way ANOVA (Figure 3), the analysis for the mean and standard deviation graph was plotted against S-PRG filler concentration and DT occlusion with the control, fresh root (7.323 ± 0.675), *0%-f* (7.323 ± 0.675), *5%-NaF* (7.087 ± 2.25), and the experimental groups, *10%-S* (6.12 ± 1.95), *20%-S* (5.877 ± 1.6), *30%-S* (4.123 ± 1.95), *40%-S* (4.54 ± 1.19), respectively. Significant differences were observed between the control group and experimental groups (*p* < 0.05). The fresh root group presented lower SD compared to all other groups.

### 3.2. Demineralization Depths on SS-OCT Image Analysis

Clear differences in SS-OCT images can be observed between unscraped and scraped parts in each specimen (Figure 4).

Furthermore, demineralization depths varied in unscraped part depending on the filler concentrations. The data on the demineralization depths are plotted in the graphs against S-PRG filler concentration, with an unscraped intact material zone and a scraped part where the material was removed with a dental evans (Figure 5 and Figure 6). The demineralization depth (Average ± SD), as scanned upon mildly air-drying for each specimen, was as follows in µm; *0%-f* unscraped (118.5 ± 31.45) and scraped (147 ± 25.19), *5%-NaF* unscraped (115.5 ± 31.45) and scraped (145 ± 22.9). In experimental groups, it was as follows: for *10%-S* unscraped (105.75 ± 36.60) and scraped (141.25 ± 30.70), *20%-S* unscraped (96.75 ± 31.96) and scraped (140.5 ± 26.19), *30%-S* unscraped (86.75 ± 27.62) and scraped (140.25 ± 25.15), and *40%-S* unscraped (84.25 ± 17.47) and scraped (141.75 ± 44.45). Remarkably, the demineralization depth was significantly less in S-PRG-incorporated groups compared to the control and *5%-NaF* groups (*p* < 0.05), although the SDs are very high and overlapped among the groups.

### 3.3. SEM Image Assessment

Representative SEM images are shown at a high magnification in Figure 7 and Figure 8. The cross-sectional view (Figure 7) and surface view (Figure 8) compared and contrasted the polymerized DV materials plugging the DTs, penetrating inside to form cylindrical tags and occluding the orifices.

From the cross-sectional view, in the control group, DV materials were unable to remain plugged, and DTs were reopened in most cases. In experimental groups, *20%-S*, *30%-S*, and *40%-S* displayed cylindrical tags deep inside many DTs. Surface-view images show the complete reopening of DT orifices in the control groups and also in *10%-S*, while partial tubule occlusion with the material at the level of the orifice can be observed in the experimental groups. A noticeable number of DT orifices remained either partially or completely plugged in *20%-S*, *30%-S*, and *40%-S*. However, decomposed DV particles are also seen at the DT orifices.

### 3.4. Elemental Analysis of DV and Surrounding Dentin

The percentile ratios of the elements in the DVs in all the specimens are shown in Table 2. For the material only group, the core value of Sr is (41.53%), present in the highest concentration, followed by F (19.96%), Al (19.63%), and Si (15.51%), compared to other S-PRG elements. A fresh dentin sample prepared from a bovine root showed high concentrations of Ca (86.35%) and P (13.08%). Higher concentrations of F (7.65%) and Na (2.06%) were detected in the *5%-NaF* sample too.

## 4. Discussion

In this study, the efficacy of the experimental DVs with different concentrations of S-PRG filler particles was evaluated. According to the results, DVs containing S-PRG filler with more than 20% wt were effective in increasing the acid resistance of the dentin surface, and the effectiveness was apparently dependent on the wt% of the S-PRG fillers.

There are variations in the source and age of collected human teeth, and the use of human teeth is associated with infection risks; therefore, bovine teeth were used in this study [28,29,30,31].

The debris and smear layer acts as a hindrance to the penetration of material inside dentinal tubules. Therefore, it was essential to remove it for the penetration of any material inside the tubules by using EDTA. The EDTA has been recommended for the removal of the organic and inorganic contents of the smear layer. The 0.5M EDTA and ethanol were used for 5 min for the effective removal of debris and microbiomes. This process made more DV materials accessible deep into the DTs. Upon the treatment with 0.5M EDTA, before application of material, the images were recorded with CLSM, and very clear DT orifices were observed with less debris, so that material can penetrate inside the DTs (data not shown).

In recent years, biofilm-based laboratory models are becoming essential for studying anti-cary therapies, because biofilms are the primary factors for cary development [22], and their presence can influence treatment outcomes [32]. Several biofilm models have previously been used to evaluate the effect of oral health care products and active ingredients on oral biofilms [32,33]. The OBR was used, as mentioned in previous studies for studying the effects of biofilm [34]. Likewise, the primary biofilm was formed in the OBR in the present study, simulating clinical dental plaque, unlike the artificial demineralization solution used previously for the evaluation of S-PRG filler containing DVs [35]. Hence, the results acquired after the biofilm challenge on the DV-treated specimens are more reliable. The incorporation of 20% to 40% S-PRG fillers in the DVs has proven to be remarkably effective, as the intrusion of bacteria and the penetration of organic acids and enzymes were inhibited, and the degradation of the material itself was resisted for 7 days. The previously published article reported that S-PRG filler containing varnishes had a superior anti-demineralization effect on root dentin, which matches the results of the present study. However, the evaluated DV with a 40% S-PRG concentration only as the objective was different.

The root surface is more susceptible to acid demineralization compared to enamel [35,36]. Products with higher fluoride (F) concentrations have been suggested for the protection of the root surface [37,38]. However, fluoride varnish films are not expected to remain longer on the surface but to be quickly removed by brushing and wear. Therefore, the effect on the coated tissues depends on the retention period and concentration of the active component, as well its release rate [1]. The technique used in the study makes it possible to observe the DV penetration depth into the DTs with a unique method for observation under CLSM, OCT, and SEM. All S-PRG-containing varnishes show some degree of penetration depth.

Upon the application of the DVs onto the dentin surface, it is expected to deliver various ions released from S-PRG components that may penetrate the dentin structures. In particular, elemental ions such as Si, F, B, and Sr may have contributed to the increase in acid resistance of the dental surface due to their remineralization effect. The incorporation of Sr with F forms an effective relationship for the prevention of caries. S-PRG fillers have been reported to facilitate the remineralization of demineralized dentin [38] and exhibit an acid buffering capacity [39]. Residual ion infiltration takes place, which may have protected the dentin demineralization in the scraped part.

As with the increased average age of the people, the number of teeth inside the oral cavity needs to be maintained for longer for a better quality of life; the development of root caries is considerably higher in older-age patients.

Even though many in vitro studies on S-PRG filler particles have been done already, to our knowledge, the present preclinical study is the first attempt to evaluate the effectiveness of different percentages of S-PRG filler in DVs against root dentin demineralization by cariogenic biofilm. An effort has been given to develop a treatment protocol for patients suffering from hypersensitivity, so the patients can relieve and prevent the demineralization of dentin. S-PRG-containing dental varnish is being developed, and the research is still in the primary in vitro stage so far; we have acquired some interesting findings, and, with these data, we have proceeded to investigate the results.

For evaluation by CLSM, after the removal of the biofilms from the surfaces, the mildly air-dried scraped parts allow the laser to penetrate better into the reopened DTs or reflect at the point occluded by the DVs. The higher and thicker peaks and deeper valleys represent that the laser penetration was greater in the control groups compared to the experimental groups, meaning more material penetration in the case of the latter. DT occlusion increased with increasing filler concentrations; those more than 20% were evidently better. However, interestingly, the SzJIS data of 40% is fractionally higher compared to 20% and 30%. This may be because flow and stickiness properties of the DV after mixing have a trend to reduce. SzJIS data analysis for the fresh root with no material and biofilm was carried out to compare the laser penetration between the differences among the groups, which definitely helps to understand the method of using the CLSM surface roughness analysis technique. Interestingly, the fresh root group presented lower SD than that of all other groups, because the root specimens were preserved intact in the refrigerator after EDTA and DiW washing with wide open DT orifices, without receiving any further processing. Conversely, DV materials were pressed into the DTs during application, and later material leached out unevenly during the biofilm test. Some bacteria or part of the biofilms could have entered too, making an obstruction for the CLSM laser. In addition, due to all these chemical and mechanical loads for 7 days, the root dentin in the scraped part was altered structurally.

The results convincingly display that *40%-S* filler varnish stays at the orifice of the dentinal tubules. However, this is not a surprise, as the heavy consistency of the varnish makes it difficult to plug the DTs and apply, which was observed during mixing too. In addition, it contains plant-extract rosin, which often forms a semi-solid mass before application. That was the reason for introducing an exclusive technique of using an evans for mixing and applying to create pressure on the mix. The considerable amount of varnish flow into the DTs and less leaching of DVs in case of *20%-S* and *30%-S* showed interesting data, even over *40%-S*. One important factor that needs to be considered is the consistency of the DVs after mixing; those thinner than *40%-S* tend to be better in terms of the mixing, application, and consistency of the varnish itself. Here, it is important to consider that the tubular direction is an important factor for varnish flow into the DTs. The flow occurs much more swiftly when tubules are parallel. However, usually there were gap-free and gap-containing regions inside the DTs. For cross-sectional imaging, the gap-containing regions were larger in the case of the control group and *40%-S*. However, *20%-S* and *30%-S* show better results. In the case of *40%-S*, the chance of material leaching is higher, as the material does not flow smoothly every time and it can proceed to secondary cary development.

The light scattering phenomenon was observed better in the mineralized part of the sample; the data shows, with an increasing percentage of filler content, in the experimental group, more light scattering occurs, which shows more material occludes the dentinal tubules.

The SS-OCT image of the mineralized lesion has less image information in the deep layer than in the surrounding decalcified area, and the lower layer of the mineralized area is darker than the surrounding area [40]. In the case of the analysis regarding the S-PRG varnishes tested in the current study, it is expected that the demineralization depth of the artificial lesions could be different if the research was performed in human dentin. However, the results of the comparison among the products would be similar. The thickness of dentin is reduced more on the scraped surface as compared to the unscraped material surface. As the purpose of partial scraping was to evaluate the effects of the experimental DVs individually as a surface coating on the same root dentin block, by comparing the coated (material remaining) part with the scraped part (removed part), it was clearly detectable by SS-OCT (could also be inspected clearly by SEM). Sound dentin under the scraped-off surface became darker with the increase in demineralization depth. On the other hand, sound dentin just under the unscraped part was brighter than that under wet conditions. Due to the 7-day acid and enzymatic attack, there was partial leaching of the DV material from the surface, mainly from the scraped part. The experimental groups showed better results against chemical or mechanical attacks. The material is localized in the DTs, light scattering occurs at the mineralized part, and the amount of light reaching the lower layer decreases. Therefore, the sound dentin appears darker in the mineralized zone compared to the demineralized zone. Occurrences of abrupt light scattering were more frequent in the case of the unscraped part, and the SDs in the graph of Figure 6 are very high and overlap among the groups. One of reasons might be that the acid and enzyme penetration through the DV coatings did not occur evenly, as it was mostly inhibited. In addition, the thicknesses of the DV coatings were not precisely uniform at all points, especially after the 7-day biofilm test. Moreover, bovine root dentin specimens were not structurally uniform, so that the demineralization of the dentin occurred unevenly. Furthermore, it might be due to the orifice size and orientation of the DTs, which were different along the scanning line of the SS-OCT.

Regarding the percentile ratio, the control group contained trace amount of elements (wt%), as expected, and compared to that, the experimental groups represented a remarkable higher ratio. Elements were detected with increasing filler content (*w*/*w*) in a higher ratio. Expectedly, high values of P and Ca were obtained, as they are the building blocks of dentin hydroxyapatite crystals. According to ionic chemistry, the ion release may have occurred from the S-PRG fillers to the surrounding dentin, which differs for each ion [39]. The release of ions from *0%-S* is almost negligible compared to the experimental groups. On the other hand, the release of Al, Si, and Sr ions is increased proportionally to the amount of S-PRG filler %wt. Dentin substrate contains P and Ca ions in a significantly higher percentile ratio. F (7.65%) and Na (2.06%) were detected at higher ratios in the *5%-NaF*-treated specimen, as it basically contained NaF filler.

As this study was designed with the view to select the best filler particle material ratio among the others, it demonstrated that *20%-S* and *30%-S* give satisfactory results within the limitations (e.g., obtaining thick and flat root dentin blocks with the more or less same number of DTs with a similar orientation and orifice size) of this study.

## 5. Conclusions

The new DVs containing more than 20% S-PRG fillers have strong potential for anti-demineralization efficacy and appear to be useful in preventing root caries and hypersensitivity, which is not essentially concentration-dependent. For better outcomes, the S-PRG filler contents need to be confined to an upper limit of less than 40%. These DVs can keep a remarkable number of plugged DTs, which could be improved through continued research and development. This study provides information useful for further studies. Research on the longevity of the new DVs through resisting degradation factors, including tooth brushing, might be interesting.

## Figures and Tables

**Figure 1 polymers-14-05015-f001:**
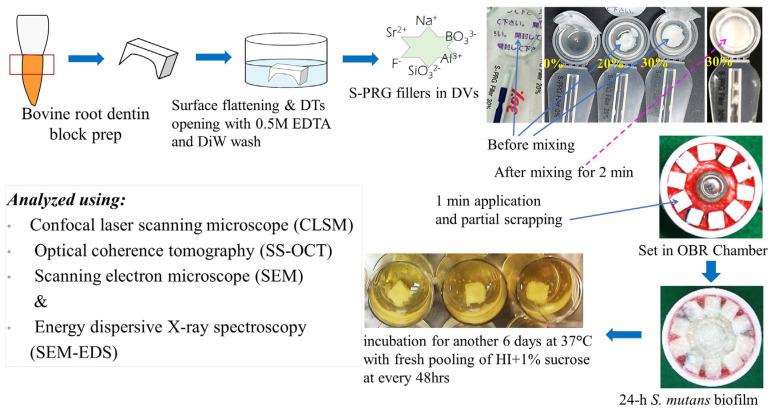
Schematic diagram shows the protocol of the study: specimen preparation, adequate mixing of active and base ingredients of each DV and application, biofilm attack for 7 days after setting of the DV materials, followed by evaluation of the effects of different DVs using CLSM, SS-OCT, SEM, and EDS. Note: each DV preparation is delivered in an air-tight sealed-pack with base and active parts together in an unmixed condition. Abbreviations: DT; dentinal tubules, DiW; de-ionized water, EDTA; ethylene diamine tetra-acetic acid, DVs; dental varnishes, OBR; oral biofilm reactor.

**Figure 2 polymers-14-05015-f002:**
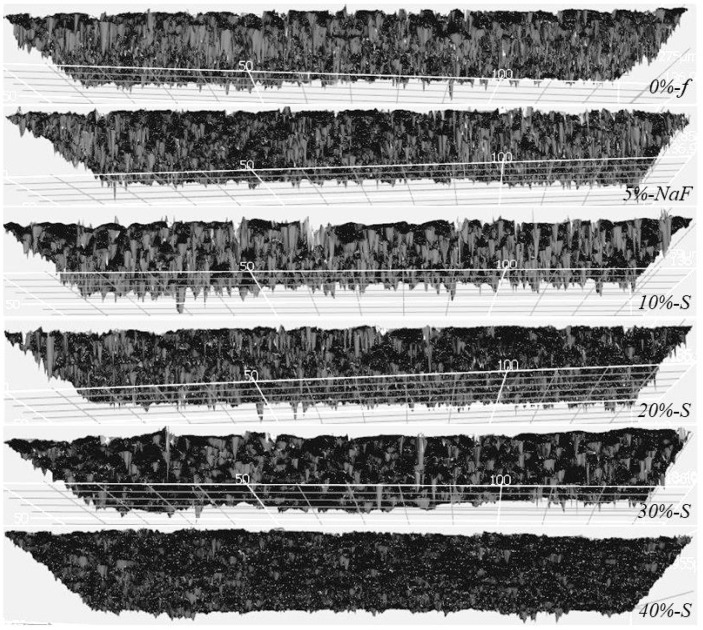
Confocal laser penetration (mountains and valleys/spikes) image of root dentin after 7-day biofilm attack. For control groups, the SzJIS is higher compared to the experimental group, showing more material occluded dentinal tubules and less laser penetration. The mean surface roughness (SzJIS; µm) of ten representative DTs for each specimen using the MultiFileAnalyzer program supplied the CLSM, a two-point analysis method. It shows SzJIS values (the average distance between five highest peaks and five lowest valleys).

**Figure 3 polymers-14-05015-f003:**
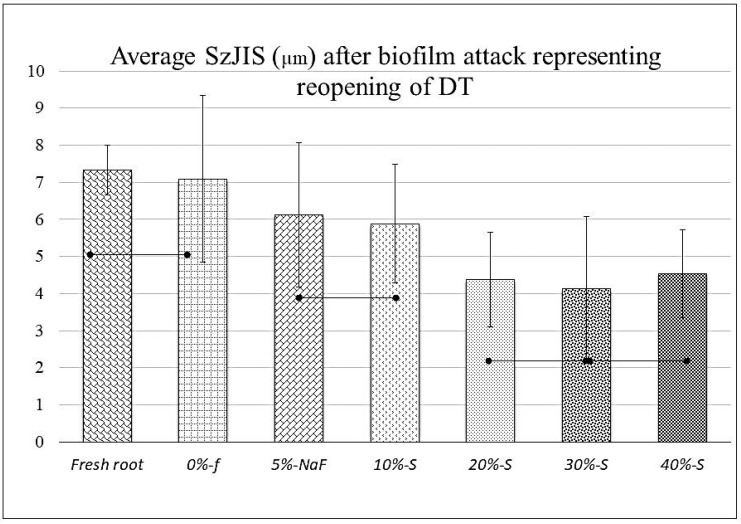
Graph of CLSM data representing depths of re-opened DTs, mean value, and standard deviation for laser penetration depths for in vitro root dentin after 7-day biofilm test for control groups (*0%-f* and *5%-Naf*) and experimental groups (*10%-S*, *20%-S*, *30%-S*, and *40%-S*)**.** Laser penetration depth (SzJIS: µm) was less significant in S-PRG-incorporated groups compared to control groups. (Dotted bars indicate no significant differences; *p* < 0.05).

**Figure 4 polymers-14-05015-f004:**
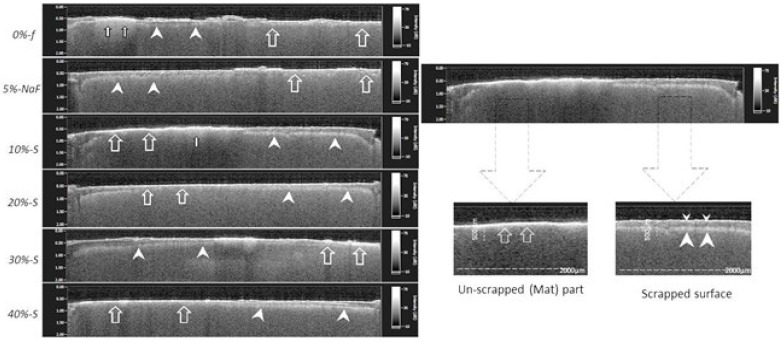
SS-OCT images displaying demineralization depths (µm) beneath DV layers; both unscraped (Mat) part (large arrows) and scraped (removed) part (arrowheads) of each specimen. Clear difference can be seen between unscraped part and scraped part; mineral loss appeared to be less in unscraped part for experimental groups, while dentin became almost transparent in craped part, indicating remarkable mineral loss for all groups. Demineralization depths were more than 140 µm on average for all materials.

**Figure 5 polymers-14-05015-f005:**
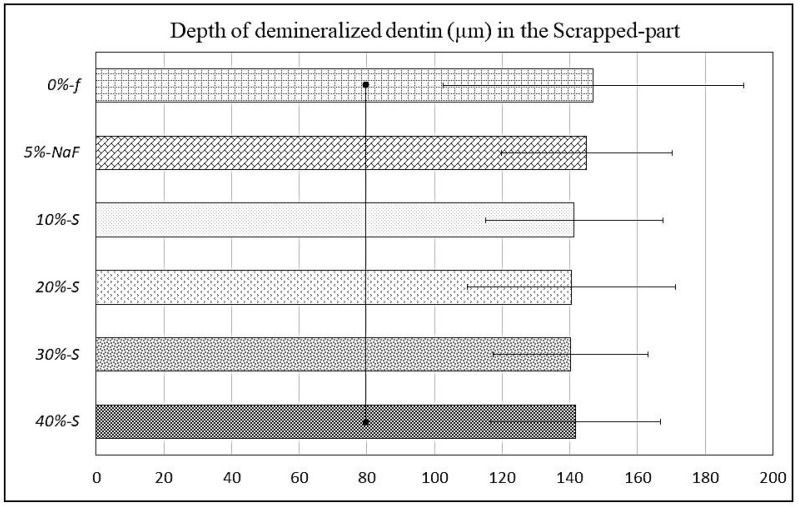
Graph of SS-OCT data representing demineralization depths (µm) beneath DV scraped (removed) part, which were more than 140 µm on average for all materials. Bars with dots indicate no significant differences (*p* < 0.05).

**Figure 6 polymers-14-05015-f006:**
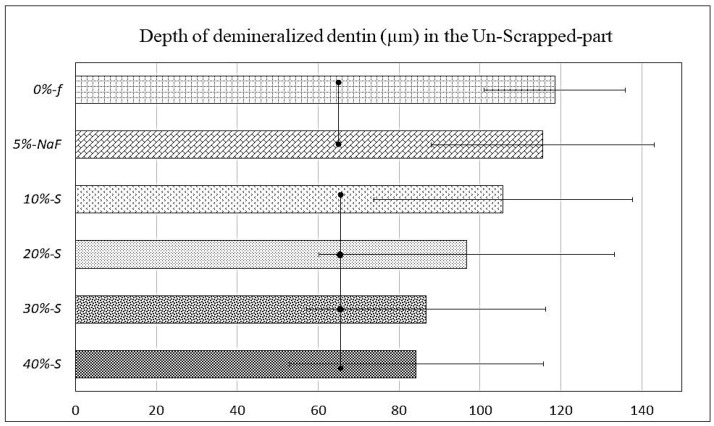
Graph of SS-OCT data representing demineralization depths (µm) beneath the DV layer, which were significantly less in all materials of the experimental group compared to the control group; *30%-S* and *40%-S* had the minimum demineralization. Bars with dots indicate no significant differences (*p* < 0.05).

**Figure 7 polymers-14-05015-f007:**
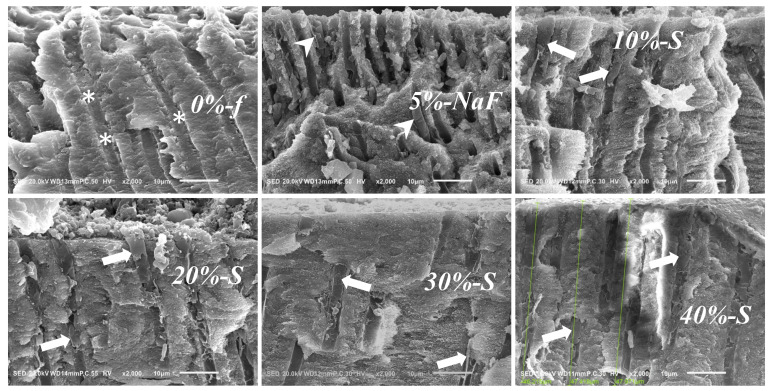
SEM images of longitudinally fractured root dentin after DV applied with pressure; original magnification: 2k. In *0-f%*, there appears to be layers of filler-less DV along the walls of the DTs (asterisks). Only a few cylindrical DVs are seen in *5%-NaF* (arrowheads). Some solidified cylindrical DV materials remain plugged in the DTs in *10%-S* (arrow). Some cylindrical DV materials remained plugged at the root surface that penetrated deep into the DTs in *20%-S* (arrow). Similar tags of DV could be seen in *30%-S* (arrow). Solidified cylindrical DV materials are seen plugged more than 47 µm deep into the DTs in *40%-S* (arrow).

**Figure 8 polymers-14-05015-f008:**
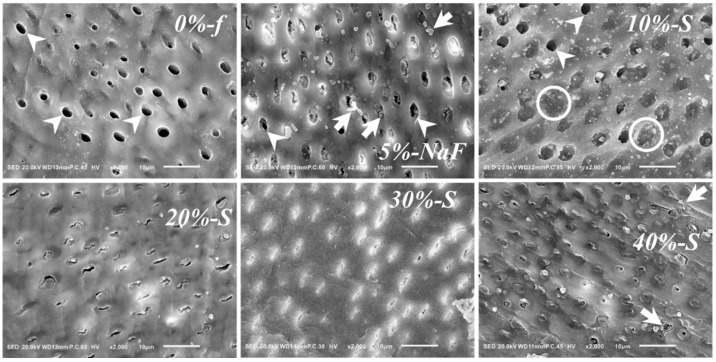
SEM images of the scraped part of root dentin surfaces after 7 days of biofilm-induced acid attack; original magnification: 2k. In *0-f%*, completely open DTs (arrowheads) are seen more, with only a few remained plugged by the filler-less material. Apparently, *5%-NaF* had decomposed DV particles (arrows) at the opening of DTs and some complete DT openings are seen. Few completely open DTs are seen in *10%-S* (arrowheads) and some are plugged with DV containing a proportionally smaller amount of fillers (circles). No completely open DTs are seen in *20%-S*; there are more partially open. In *30%-S*, only crack-like DT openings are seen. Most DTs remain completely occluded in *40%-S*; decomposed DV particles are also seen at the DT opening (arrows).

**Table 1 polymers-14-05015-t001:** DV preparations with their group distribution, lot numbers, and composition are shown. Abbreviations: TEGDMA; triethyleneglycol dimethacrylate, DW; distilled water, bis-MPEPP; 2,2-bis(4-methacryloxy polyethoxyphenyl) propane.

Materials	Group	Lot Number	Composition
0% S-PRG (*w*/*w*)	*0%-f*	180130	Same active ingredients as shown below and base component without any fillers
5% NaF (*w*/*w*)	*5%-NaF*	180306	Same active ingredients as shown below and base with 5% NaF filler only
10% S-PRG (*w*/*w*)	*10%-S*	180130	Same active ingredients as shown below and base with 10% S-PRG fillers
20% S-PRG (*w*/*w*)	*20%-S*	180130	Same active ingredients as shown below and base with 20% S-PRG fillers
30% S-PRG (*w*/*w*)	*30%-S*	180130	Same active ingredients as shown below and vase with 30% S-PRG fillers
40% S-PRG (*w*/*w*)	*40%-S*	180130	Same active ingredients as shown below and vase with 40% S-PRG fillers
**Active:** Phosphonic acid monomer, methacrylic acid monomer, bis-MPEPP, carboxylic monomer, TEGDMA, initiator, others **Base:** S-PRG filler (mean diameter, 3.0 μm), DW, methacrylic acid monomer, others			

**Table 2 polymers-14-05015-t002:** Percentile distribution of filler elements inside and around DTs after application in relation to Ca and P obtained on analysis by an EDS. Abbreviations: DTs; dentinal tubules, B; boron, F; fluoride, Na; sodium, Al; aluminum; Si; silicon, P; phosphorous, Ca; calcium, and Sr; strontium.

Elements	B	F	Na	Al	Si	P	Ca	Sr
Dentin	0	0.09	0.03	0.26	0.08	13.08	86.35	0.1
*0%-f*	0	0.62	0.13	0.27	0.01	29.42	69.51	0.03
*5%-NaF*	0	7.65	2.06	0.34	0.05	30.91	58.94	0.04
*10%-S*	0	6.07	2.19	0.4	0	29.35	61.31	0.67
*20%-S*	0	5.45	1.82	1.22	0	30.66	59.8	1.04
*30%-S*	0	8.17	1.57	0.99	0.93	30.97	57.10	0.27
*40%-S*	0	6.24	1.57	1.29	0.02	29.25	59.54	2.1
Base material	0	19.96	3.37	19.63	15.51	0	0	41.53

## Data Availability

The data presented in this study are available on request from the corresponding author.

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
