# Peer review of "Optimal Surface Pre-Reacted Glass Filler Ratio in a Dental Varnish Effective for Inhibition of Biofilm-Induced Root Dentin Demineralization"

_polymers, 2022, doi:10.3390/polym14225015_

Round 1

Reviewer 1 Report

Please revise the text of your manuscript considering the items below.

1) Grammar: some sentences have problems that impair the understanding of the manuscript.

2) Standardization: please make sure you provide the meaning of all abbreviations used; after that, please use the abbreviations. Sometimes, the abbreviation and the whole name are mixed along the text.

3) Figure 1 and its legend could be improved.

4) The results (values) presented in the graphs/tables are also presented within the text.

5) The paragraph about “Statistical analysis” needs to be revised.

6) In the list of references, some references are incomplete.

7) Among the references, there are eight references before 2000; one reference 2020 and one 2021.

Author Response

Reviewer advised: [Please revise the text of your manuscript considering the items below. ]

Thank you very much for kindly reviewing our manuscript and your valuable comments. We sincerely appreciate your kind help in improving the quality and presentation of the manuscript.

  • Grammar: some sentences have problems that impair the understanding of the manuscript.

Checked and corrected in the revised manuscript. Thank you very much.

2) Standardization: please make sure you provide the meaning of all abbreviations used; after that, please use the abbreviations. Sometimes, the abbreviation and the whole name are mixed along the text.

Revised as advised.

3) Figure 1 and its legend could be improved.

Revised as advised, please check the revised manuscript.

4) The results (values) presented in the graphs/tables are also presented within the text.

You have rightly pointed that out; we thought keeping as it is might be helpful for the readers!

5) The paragraph about “Statistical analysis” needs to be revised.

Revised to minimize errors. Thank you very much.

6) In the list of references, some references are incomplete.

Corrected, please check.

7) Among the references, there are eight references before 2000; one reference 2020 and one 2021.

References before 2000 removed; replaced with newer references.

Reviewer 2 Report

The manuscript of polymers-1982019 entitled “Optimal surface pre-reacted glass filler ratio in a dental-varnish 2 for inhibiting cariogenic-biofilm induced root-dentin deminer-3 alization and plaguing dentinal tubules”. The manuscript in its current form lacked a strong justification for the novelty of the study. Take note that simply having the thought of “However, studies on the general health and dental effects including fluorosis and the safety of fluoride [14] described that only fluoride-releasing ability is not enough...” would not warrant sufficient justification of the novelty. Unfortunately, I would recommend publication for Polymers.

Other comments for consideration for the next submission:

1.       Proofread the whole manuscript to minimize text or grammatical errors.

2.       In the last paragraph of the introduction section, please underscore the research gap and novelty of the study.

3.       Please underscore the scientific value-added of your paper in your abstract and introduction.

4.       Please make sure your conclusions section underscores the scientific value-added of your paper.

5.       There should be no lumping of references. This can signify a bad literature review. Please separate the citation in its own sentence to highlight each work properly.

6.       Update your references accordingly. Cite references in the last 3 years for future projection discussions.

7.       Upgrade the quality of all your figures. A good image should still be clear even if you “drastically zoom in” for the figure.

8.       A critical comparison with previous studies is also missing.

9.   The authors are advised to write the conclusions in a comprehensive way and should contain key values, suitability of the applied method, the major findings, contributions, and possible future outcomes.

Author Response

Reviewer wrote: [Other comments for consideration for the next submission:]

We would like to express our sincere gratitude for reviewing our manuscript.

Thank you very much for your valuable comments and we sincerely appreciate your kind help in improving the quality and presentation of the manuscript.

  1. Proofread the whole manuscript to minimize text or grammatical errors.

We have proofread the whole manuscript as suggested to minimize text or grammatical errors, please check.

  1. In the last paragraph of the introduction section, please underscore the research gap and novelty of the study.

Revised as advised. Thank you very much for your valuable comment.

  1. Please underscore the scientific value-added of your paper in your abstract and introduction.

Revised as advised. Thank you very much

  1. Please make sure your conclusions section underscores the scientific value-added of your paper.

Revised as advised. We appreciate your kind help by providing valuable comments.

  1. There should be no lumping of references. This can signify a bad literature review. Please separate the citation in its own sentence to highlight each work properly.

Revised as advised as much as we could.

  1. Update your references accordingly. Cite references in the last 3 years for future projection discussions.

Updated references. Thank you very much.

  1.      Upgrade the quality of all your figures. A good image should still be clear even if you “drastically zoom in” for the figure.

Thank you, we have upgraded and provided/ uploaded original PowerPoint files with much better resolution and a high magnification SEM image in Graphical Abstract.

  1. A critical comparison with previous studies is also missing.

 Included in the revised manuscript (also considered comments from other reviewers).

  1. The authors are advised to write the conclusions in a comprehensive way and should contain key values, suitability of the applied method, the major findings, contributions, and possible future outcomes.

Revised as advised. Thank you very much. We sincerely appreciate your kind help and cooperation.

Reviewer 3 Report

The study seems very interesting and may add potential information to the current literature, however the authors should address the following comments to improve the quality and presentation of the manuscript:

- The title is too long and authors may shorten it while keeping the same theme.

- The abstract should include a short statement on the existing research gap and outstanding research question (please maintain the word limit).

- Line 42 and 43: The authors cited one review paper and referred to it by "several articles". Authors may refer to it as a review paper that involved many articles.

- Please mention the research hypothesis/hypotheses at the end of the introduction section and authors stand to accept or reject them in the discussion section.

- What was the storage media for the bovine teeth (line 85)?

- Figure 1: the images on the right side showing the specimens storage and treatment are not clear, please consider modification of size and resolution for clarity.

- "& Milli-Q for 5 minutes" the authors should clarify the procedure that was performed here.

- Table 1: the word "ingradients" should be replaced with "ingredients".

- Line 112: Please mention the method used for cleaning and duration.

- Line 116: Please explain the term "dental evans".

- What was the purpose of partial scrapping of the applied material? Please reflect that in the manuscript.

- In figure 3: the SD seems higher in experimental groups as compared to the fresh root group. Please elaborate on that in the results and discussion sections.

- In figure 6: The SD is very high and overlapped among the groups. Please elaborate on this observation.

- The authors should mention the study limitations and directions for future research at the end of the discussion section.

- Conclusion section should be a separate section and can be summarized in bullets for better presentation.

Author Response

Reviewer wrote: [The study seems very interesting and may add potential information to the current literature, however the authors should address the following comments to improve the quality and presentation of the manuscript:]

We would to express our sincere gratitude for reviewing our manuscript and kindly providing valuable comments. We sincerely appreciate your kind help in improving the quality and presentation of the manuscript.

- The title is too long and authors may shorten it while keeping the same theme.

Shortened as advised. [Optimal surface pre-reacted glass filler ratio in a dental-varnish effective for inhibition of biofilm induced root-dentin demineralization]

- The abstract should include a short statement on the existing research gap and outstanding research question (please maintain the word limit).

Revised as advised maintaining the word limit. Than you very much for your kind advise.  

- Line 42 and 43: The authors cited one review paper and referred to it by "several articles". Authors may refer to it as a review paper that involved many articles.

Corrected as advised.

- Please mention the research hypothesis/hypotheses at the end of the introduction section and authors stand to accept or reject them in the discussion section.

Mentioned the research hypothesis/hypotheses at the end of the introduction section and written our stand in Discussion and Conclusions parts.

- What was the storage media for the bovine teeth (line 85)?

No storage media was used at this stage. In all other experiments in our lab (articles published before) use same storage procedure. 

- Figure 1: the images on the right side showing the specimens storage and treatment are not clear, please consider modification of size and resolution for clarity.

Modified the size and resolution as advised (also provided the original PowerPoint file to the editorial office).

- "& Milli-Q for 5 minutes" the authors should clarify the procedure that was performed here.

Revised to clarify the procedure as advised, thank you.

- Table 1: the word "ingradients" should be replaced with "ingredients".

Replaced as advised. Thank you.

- Line 112: Please mention the method used for cleaning and duration.

Mentioned in the revised manuscript, please check.

- Line 116: Please explain the term "dental evans".

A dental-lab instrument, one end with a small stainless steel knife and a spoon one the other.

Please check the URL~   https://www.dentech.co.jp/en/post_product/evans/

- What was the purpose of partial scrapping of the applied material? Please reflect that in the manuscript.

The purpose of partial scrapping was to evaluate effects of the experimental DVs individually as a surface-coat on the same root-dentin block; by comparing coated (material remaining) part with the scrapped part (removed part) was clearly detectable by SS-OCT (could also be inspected clearly by SEM) .

- In figure 3: the SD seems higher in experimental groups as compared to the fresh root group. Please elaborate on that in the results and discussion sections.

In reverse, the fresh root group presented lower SD than that of others all groups because the root specimens preserved in the refrigerator intact after EDTA and DiW wash with wide open DT orifices without receiving any further processing. In reverse, as DV materials were pressed into the DTs while apply and later may have leached out unevenly during biofilm challenge. Some bacteria or part of the biofilms could have entered too. In addition, due to all these chemical and mechanical loads for 7-days the root-dentin got damaged structurally as well. (Elaborated in the results and discussion sections).        

- In figure 6: The SD is very high and overlapped among the groups. Please elaborate on this observation.

Occurrences of abrupt light scattering of SS-OCT were more in case un-scrapped part and the SDs in the graph of Fig. 6 are very high and overlapped among the groups. One of reasons might be acid and enzyme penetration through the DV-coats didn’t occur evenly as was inhibited mostly. Also, the thicknesses of the DV-coats were not precisely uniform at all points, especially after 7-day biofilm challenge. Moreover, bovine root-dentin specimens were not structurally uniform, so that demineralization of dentin occurred unevenly. Also, might be due to orifice-size and orientation of the DTs were different along the scanning line of SS-OCT. Elaborated in the revised manuscript, please check.

- The authors should mention the study limitations and directions for future research at the end of the discussion section.

Thank you again. Completely agreeing with the comment we have rewritten the paragraph.

- Conclusion section should be a separate section and can be summarized in bullets for better presentation.

Prepared as advised, thank you. We sincerely appreciate your kind help and cooperation.

Round 2

Reviewer 2 Report

The authors have now properly addressed all of my comments.